# Influence of Polymer Viscoelasticity on Microscopic Remaining Oil Production

**DOI:** 10.3390/polym14050940

**Published:** 2022-02-26

**Authors:** Yiqun Yan, Lihui Wang, Guoqiang Sang, Xu Han

**Affiliations:** 1Research Institute of Petroleum Exploration & Development, PetroChina, Beijing 100083, China; sgqminer@petrochina.com.cn (G.S.); dq_hanxu@hotmail.com (X.H.); 2Postdoctoral Research Workstation of Daqing Oilfield, Daqing 163458, China; 3Postdoctoral Research Mobile Station of Northeast Petroleum University, Daqing 163318, China; 4Exploration and Development Research Institute of Daqing Oilfield Co., Ltd., Daqing 163712, China

**Keywords:** polymer, viscoelasticity, microscopic remaining oil, force analysis, oil displacement effect

## Abstract

To investigate the impact of polymer viscoelasticity on microscopic remaining oil production, this study used microscopic oil displacement visualisation technology, numerical simulations in PolyFlow software, and core seepage experiments to study the viscoelasticity of polymers and their elastic effects in porous media. We analysed the forces affecting the microscopic remaining oil in different directions, and the influence of polymer viscoelasticity on the displacement efficiency of microscopic remaining oil. The results demonstrated that the greater the viscosity of the polymer, the greater the deformation and the higher the elasticity proportion. In addition, during the creep recovery experiment at low speed, the polymer solution was mainly viscous, while at high speed it was mainly elastic. When the polymer viscosity reached 125 mPa·s, the core effective permeability reached 100 × 10^−3^ μm^2^, and the equivalent shear rate exceeded 1000 s^−1^, the polymer exhibited an elastic effect in the porous medium and the viscosity curve displayed an ‘upward’ phenomenon. Moreover, the difference in the normal deviatoric stress and horizontal stress acting on the microscopic remaining oil increased exponentially as the viscosity of the polymer increased. The greater the viscosity of the polymer, the greater the remaining oil deformation. During the microscopic visualisation flooding experiment, the viscosity of the polymer, the scope of the mainstream line, and the recovery factor all increased. The scope of spread in the shunt line area significantly increased, but the recovery factor was significantly lower than that in the mainstream line. The amount of remaining oil in the unaffected microscopic area also decreased.

## 1. Introduction

Polymer flooding can greatly improve the oil recovery from waterflooding development oilfields because the higher viscosity of polymer solutions can effectively improve the oil–water mobility ratio, alleviate the conflicts between layers, improve the fluid absorption profile, and enhance displacement [1,2,3]. Moreover, the viscoelastic effect of the polymer solution can effectively mobilise the remaining oil after waterflooding, improve the oil displacement efficiency, and significantly increase the oil recovery [4,5,6,7,8]. Presently, the Daqing oilfield is the world’s largest polymer flooding application block. The annual polymer flooding oil production accounts for more than 30% of the total oil production. Polymer flooding has great potential for increasing oil production [9,10,11,12,13].

Although polymer flooding has been utilised for more than 40 years, there is still controversy regarding the mechanism of the viscoelastic effect of polymers in the process of percolation [14,15,16,17,18]. The focus of previous studies has been on the impact of polymer viscoelasticity on the oil displacement effect and the critical conditions under which the polymer solution can exhibit elastic effects. When the polymer viscosity is low, the viscoelastic effect cannot occur, and when the viscosity is too high, the viscoelastic effect is also impaired. The injection performance and equivalent shear rate in porous media also have significant impacts on viscoelastic performance [19,20,21,22]. Therefore, the rational use of the viscoelasticity of a polymer can maximise the oil displacement efficiency at a relatively low polymer dosage. This is of immense significance for the efficient development of oil reservoirs [23,24,25].

To describe the impact of the polymer viscoelastic effect on the microscopic oil displacement effect more accurately, scholars worldwide have used different methods and theories to study the viscoelasticity of polymers [26,27,28]. The elastic effect of a compound solution is mainly investigated through experiments that have attracted substantial attention. Gogarty and Levy studied the flow of viscoelastic polymer solutions in a porous medium [29,30,31,32,33]. Several comparison schemes confirmed that the total pressure drop in the flow is affected by the viscosity and elasticity. Furthermore, the Darcy equation has been modified to separately calculate the fluid. The viscosity and elastic pressure drop render this formula suitable for elastic non-Newtonian fluids. Doughty theoretically studied the four constitutive models of Oldroyd, Pao, Bogue, and BKZ to calculate linear rheological parameters. By fitting the rheological experimental data and theoretical calculations, the BKZ constitutive model has been used to analyse the rheology of the polymer solutions, as described below. Garrouch used artificial glass bead cores and Bailey cores were used for percolation experiments, which proved that a polymer solution is not only viscous but also elastic [34,35,36,37]. When using the Debra number to describe viscoelastic flow, a dimensional viscous number has been proposed to describe the viscoelastic flow of a polymer solution, which has been verified by fitting based on experimental data [38]. The above studies have conducted innovative research on the theory of polymer seepage in porous media. However, the analysis of polymer viscoelasticity is inadequate, with only macroscopic theoretical analysis and little experimental verification. The effect of different viscoelastic polymers on microscopic remaining oil production is still unclear, and PolyFlow software has not been used to numerically simulate the force of different viscoelastic polymers on microscopic remaining oil [39]. It is unfeasible to comprehensively analyse the polymer from both macroscopic and microscopic aspects to determine the effects of viscoelasticity on oil displacement.

The purpose of this study was to apply microscopic visualisation oil displacement technology combined with numerical simulation technology and an indoor core seepage experiment to comprehensively analyse the influence of polymer viscoelasticity on the microscopic oil displacement effect. Therefore, we chose the polymer applied in the Daqing oilfield as the research object. Through a creep recovery experiment and an evaluation of rheological properties, the viscosity and elasticity of polymer solutions with different viscosities were quantitatively characterised. A core experiment was used to describe the elastic effect of the polymer solution based on changes in the polymer solution’s viscosity in the porous medium. PolyFlow software was used to calculate the normal deviatoric stress and horizontal stress difference of the polymer solutions with different viscosities in simplified micropores. The dynamic images collected by microscopic visualisation of oil displacement technology illustrated the influence of different viscoelastic polymers on the microscopic oil displacement effect.

## 2. Materials and Methods

### 2.1. Materials

Chemicals: Partially hydrolysed polyacrylamide with a molecular weight of 25 million and an effective content of 90% was produced by Daqing Refining & Chemical Company (Daqing, China).

Experimental core: The experimental cores included natural cores with a length of 10 cm, diameter of 2.5 cm, and effective permeability of 50, 100, or 150 × 10^−3^ μm^2^.

Microscopic visualisation model of glass etching: Photochemical etching technology was used to create a photoetched glass model. A real pore structure photo of the cast sheet was placed on the glass coated with photosensitive material, and the pore outline pattern was copied onto the glass after exposure. The exposed glass template was then treated with hydrofluoric acid to show the impression of the pore structure, and a cover plate was added and sintered at a high temperature. The model, with a size of 40 mm × 40 mm, was developed by a professional company.

Oil and water: Simulated formation water: The NaCl content of clean water was 950 mg/L, while that of sewage was 4500 mg/L. All systems used clean water to prepare the mother liquor and sewage to dilute the target liquid and model saturated water. Sewage was also used in the displacement experiments. Experimental oil: Simulated oil prepared at a certain ratio between crude oil and kerosene had a viscosity of 10 mPa·s at 45 °C.

### 2.2. Experimental Methodology

#### 2.2.1. Core Flow Experiment

A HAKERS150 rheometer was used to detect the rheology of the polymer solution, and the elastic effect of the polymer in the porous medium was characterised based on the results of the indoor core experiment. Cores with effective permeabilities of 50, 100, and 150 × 10^−3^ μm^2^ were saturated with simulated formation water. The polymer mother liquor was prepared with clear water with an NaCl content of 950 mg/L. Simulated formation water with an NaCl content of 4500 mg/L was used to prepare the target solution, and polymer solutions with viscosities of 40, 70, 125, and 215 mPa·s were prepared. Flow experiments were then initiated for cores with different permeabilities at displacement rates of 8, 16, 32, 64, and 128 mL/h. Utilizing this formula, the displacement rate was converted into the shear rate of the polymer solution in the porous medium. At the outlet end, the displacement fluid was used to test its viscosity, and the viscoelastic characteristics of polymer migration in the porous medium were described according to the change in the viscosity of the solution.

#### 2.2.2. Analysis of Force and Deformation of Microscopic Remaining Oil

To perform numerical simulation calculations and analyse the flow field, it is necessary to simplify the microchannels with complex shapes in the core. In this study, the flow channel and remaining oil model in the two-dimensional flow field after the flow channel were simplified. The physical model was abstracted and simplified. It was assumed that the displacement fluid flowed between two infinite parallel plates with only one inlet and one outlet. It is assumed that the length of the flow channel in Figure 1 is L = 100 μm, the width of the flow channel is H = 20 μm, and the remaining oil size is set according to the simulation conditions. The displacement fluid flows under the effect of the pressure difference in the x-direction; the effect of gravity is neglected, and the boundary layer is ignored. Presuming that the flow channel is smooth, the interface between the displacement phase and oil is a moving boundary that changes with time. When calculating the microscopic remaining oil deformation, one can assume that three-phase contact points A and B are fixed [40,41,42]. 

Basic equation.

The continuity equation is an expression of the law of conservation of mass for a system where a fluid is applied to the control body. Its differential form is as follows:(1)$$\frac{d\rho }{dt}+\rho divu=0$$
where ρ represents the density of the fluid (g/cm^3^) and u represents the velocity of the fluid particles (m/s).

As both the continuous phase (displacement fluid) and dispersed phase (oil) are incompressible fluids, *ρ* = C. Thus, the continuity equation is deformed as follows:(2)∇⋅u=0

As the interface is a moving boundary, the effect of interfacial tension is considered in the equation of motion. The equation of motion is expressed as follows:(3)$$\frac{\partial (\rho u)}{\partial t}+\nabla \cdot (\rho u)=\nabla \cdot T+\int_{\partial B}dx_{B}\kappa n\sigma \delta (x-x_{B})$$
where σ represents the interfacial tension between the polymer solution and the oil (mN/m; considered as a constant value in this study); ∂B represents the remaining oil surface including point x_B_; κ represents the curvature of the remaining oil surface; *n* represents the external method of the remaining oil surface to the unit vector; *δ*(x−x_B_) represents the second-order δ function; and T represents the stress tensor [43].

In this study, the upper-satellite Maxwell constitutive equation was used to describe the seepage characteristics of a displacement fluid in simplified pores. The constitutive equation used was as follows:(4)$$\tau^{ik}+\lambda \overset{\nabla }{\tau^{ik}}=\eta_{0}A^{ik}$$
where λ is the relaxation time (s), $\eta_{0}$ is the zero-shear viscosity (Pa·s), *A* is the first-order Rivlin–Ericksen deformation tensor, and $\overset{\nabla }{\tau^{ik}}$ is the upper satellite derivative.

In the constitutive equation:(5)$$A^{ik}=\sqrt{g^{kk}g^{ii}}\frac{A_{ik}}{\sqrt{g_{kk}g_{ii}}}=A_{ik}=v_{i,j}+v_{j,i}=\frac{\partial v_{i}}{\partial x_{j}}+\frac{\partial v_{j}}{\partial x_{i}}$$

The upper satellite derivative is:(6)$$\overset{\nabla }{\tau^{ik}}=\frac{\partial \tau^{ik}}{\partial t}+v^{m}\frac{\partial \tau^{ik}}{\partial x^{m}}-\tau^{mk}\frac{\partial v^{i}}{\partial x^{m}}-\tau^{im}\frac{\partial v^{k}}{\partial x^{m}}$$

2.Numerical solution method.

By solving the flow equation, the stress and horizontal stress difference of the remaining oil can be calculated. When describing the deformation of the microscopic remaining oil, assuming that the remaining oil is in a static state, calculating the flow field of the remaining oil during its displacement by the polymer solution is a steady problem. However, when the remaining oil deforms under the continuous displacement of the polymer solution, it is an unsteady problem, which involves the deformation and reconstruction of the liquid–liquid interface. The polymer solution has complex rheological properties and flow equations. Thus, it is necessary to consider both the interfacial tension and time-varying properties.

As the interface between the displacement fluid and the remaining oil changes dynamically during the displacement process, the deformation of the oil–water interface must be captured by an interface tracking method. In this study, the Lagrange interface-tracking method was used to track and reconstruct a two-phase interface. During this method, the interface position of the closed curve is described by parameters. The dynamic interfacial deformation of oil and water changes with the movement of the closed curve. Mark points were set on the closed curve, which were used to track the position change of the interface. Lagrange multipliers can accurately determine the changes in the moving interface and prevent the spread of the value [44].

3.Boundary conditions.

Assuming that the remaining oil is stationary, the calculation of the deviatoric stress and horizontal stress difference acting on the remaining oil is a steady-state problem, and only boundary conditions need to be considered. The inlet and outlet flows were used in this study, and the rest were treated as wall surfaces. The deformation of the remaining oil under the action of a displacement fluid is an unsteady-state problem. At the interface between the displacement fluid and remaining oil, the interface conditions are satisfied, and the tangential stress and velocity are continuous. In this study, it was assumed that the remaining oil and rock contact points A and B do not move. Thus, the remaining microscopic oil does not initially deform.

#### 2.2.3. Microscopic Visualisation Flooding Experiment

To study the influence of polymer viscosity on the microscopic oil displacement effect, polymer solutions with different viscosities were prepared to conduct microscopic visualisation experiments, and a Hacker rheometer was used to test the viscosity. The viscosity–concentration relationship is shown in Table 1.

The polymer solution was configured according to Table 1, and the experimental steps were as follows: (1) The microscopic visualisation model was dried and weighed. Then, a vacuum pump was used to vacuum the saturated simulated oil, the weight of the model after saturation with oil was determined, and the pore volume and initial oil saturation was calculated based on the mass difference before and after the model was saturated with oil. (2) After saturation with oil, the model was placed in a 45 °C thermostat for 48 h. (3) After removing the model from the thermostat, the water drive was tested first, and the simulated formation water was used for constant rate displacement (0.03 mL/h). When the oil was displaced to the outlet end of the model and no oil was produced, the microscopic visual image analysis system was used to calculate the recovery factor in the model at this time. (4) After water flooding, flooding was conducted with the polymer with a viscosity of 40 mPa·s, and no oil was produced after displacement to the outlet. The recovery factor in the model after polymer flooding was then calculated. The polymer concentration viscosity was 70 mPa·s and the viscosity was 125 mPa·s. The above steps were repeated for oil displacement experiment with a polymer viscosity of 215 mPa·s. (5) After polymer flooding, subsequent water flooding was performed until no oil was produced at the outlet end. The experimental setup is illustrated in Figure 2.

## 3. Results and Discussion

### 3.1. Analysis of Viscoelasticity of the Polymer Solution

#### 3.1.1. Polymer Creep Recovery Performance

A HAKERS150 rheometer was used to perform dynamic experiments on the creep recovery performance at 45 °C. Changes in the deformation of the polymer solution over time and the changes in viscoelasticity were studied, and the proportions of elasticity and viscosity in the polymer solution were analysed, as shown in Figure 3. In the creep recovery experiment, a constant stress was applied to the sample at t = t_0_. At t = t_1_, the stress was removed, and the strain changed with time during the process. In the creep stage, the applied stress produces a transient response, the viscoelasticity of each substance has a combined effect, and their specific contributions cannot be clearly separated [45]. After the applied stress was released, it entered the recovery state. The advantage of the recovery phase is that the percentage of the total strain can be decomposed into a permanently maintained viscous part and recoverable elastic part. The ratio of the strain contributed by the elastic part and that contributed by the viscous part to the total strain reflect the elasticity and viscosity of the viscoelastic solution, respectively. In the creep phase, the deformation increased with time. In the recovery phase, as the time increased, the deformation first decreased sharply and then stabilised. As the viscosity of the polymer solution increased, the proportion of the elastic part increased.

#### 3.1.2. Quantitative Characterisation of Polymer Viscoelasticity

As shown from the change curve of the storage modulus (*G′*) and dissipation modulus (*G″*) of polymer solutions with different viscosities with angular velocity, as the angular velocity increased, the storage modulus and energy dissipation modulus increased. In addition, as the viscosity of the solution increased, the storage modulus and energy dissipation modulus increased, the number of molecules per unit volume increased, and the ability of molecules to attract and entangle with each other increased. This led to an increase in the strength of viscoelasticity. The storage and dissipation moduli of each viscous polymer solution exhibited an intersection. The intersection point moved to the left; *G″* was greater than *G′* before the intersection point, and *G′* was greater than *G″* after the intersection point. At low angular velocities, the polymer solutions were mainly viscous, whereas at high angular velocities, they were mainly elastic [46]. As shown in Figure 4.

Figure 5 shows the storage and energy dissipation moduli of polymer solutions with viscosities of 40, 70, 125, and 215 mPa·s at shear rates of 0.06283 and 6.283 s^−1^. According to the results of the dynamic experiment, when the shear rate was low (0.06283 s^−1^), the proportions of the elastic and viscous components remained relatively unchanged at 44.75% and 54.98%, respectively. Under the formation shear rate (6.283 s^−1^), the proportion of elasticity far exceeded that of the viscous part, and with an increase in the mass concentration of the solution, the proportion of the elastic part increased. However, the magnitude of the increase gradually decreased. This phenomenon indicates that the elasticity of the solution is not negligible when the polymer solution flows into the porous medium. The greater the viscosity of the polymer solution, the more significant the elastic effect. The results of the creep recovery experiment under low stress conditions and the dynamic experiment under low shear rate conditions were slightly different because the two measurement methods are based on different principles. In the creep recovery experiment, the elastic and viscous components were expressed by the relative magnitude of the contribution of the elastic and viscous properties to the strain. The ratio of the strain contributed by the elastic properties and that contributed by the viscous properties to the total strain directly reflects the elasticity and viscosity of the viscoelastic solution. The storage and dissipation moduli were used to characterise the viscoelasticity of the solution based on the relationship between the force and corresponding deformation. When the solution viscosity reached 125 mPa·s, the results of the two methods were consistent [47].

To understand the structure of partially hydrolysed polyacrylamide, the morphology of the film formed by titrating the above polymer solution onto a glass slide was observed and analysed using an atomic force microscope. Figure 6 shows the structural morphology of the polymer solutions with different viscosities; the scanning range was 20 μm × 20 μm. Although the polymer segments had a certain probability of collision in the solution, they could not be stably bonded by van der Waals forces owing to the action of the solvent and thermal movement. After the polymer solution was dropped onto the surface of the glass slide, the polyacrylamide molecules moved freely to form the most stable distribution. The distribution of polyacrylamide was fixed until the water evaporated. Polyacrylamide does not stretch and advance individually during movement, although multiple molecules can gather into small particles to move. From the atomic force scanning images of polymer solutions with different viscosities, when the viscosity was 40 mPa·s, the particles were almost isolated and did not have an orderly distribution, indicating that the viscoelasticity of the solution was very low. When the viscosity was 70 mPa·s, the distance between particles decreased and aggregates formed; however, the distribution of the aggregates was uneven, and the distance between them was still relatively large. Compared to those in a polymer solution with a viscosity of 40 mPa·s, the number of molecules increased, and the viscoelasticity increased. When the viscosity increased to 125 mPa·s, an ordered arrangement of polyacrylamide polymer chains easily formed, and a relatively regular network structure developed between the particles. The structure of the solution at this point was ideal and the proportion of elasticity reached a maximum. When the viscosity of the solution continued to increase to 215 mPa·s, the probability of intermolecular collisions increased; thus, a self-similar tree structure formed between particles, and the increase in the number of molecules led to an increase in the overall viscoelasticity of the solution [48].

#### 3.1.3. Elastic Effect of Polymers in Porous Media

When a polymer flows into a pore, it is subjected to shearing and stretching by the porous medium, and the polymer solution exhibits viscoelasticity when the viscosity and shear rate reach a certain level. The different displacement rates allow the calculation of the shear rate of the polymer solution during its formation. The viscosity curves of polymer solutions with different viscosities after shearing in core samples with different permeabilities are shown in Figure 7. In the core with a permeability of 50 × 10^−3^ μm^2^, as the shear rate increased, the viscosity of the polymer solutions with different viscosities gradually decreased and did not exhibit viscoelasticity. This is mainly because the permeability and porosity were low, while the shear rate of the polymer solution in the core was high, which significantly damaged the polymer molecular structure. Consequently, the viscosity of the polymer gradually decreased, thus not exhibiting viscoelasticity. In the core with a permeability of 100 × 10^−3^ μm^2^, when the polymer viscosity was 40 and 70 mPa·s, as the shear rate increased, the viscosity gradually decreased. When the viscosity of the polymer solution reached 125 mPa·s and the shear rate exceeded 1000 s^−1^, the polymer solution after core shearing exhibited viscoelasticity, and the viscosity curve exhibited an ‘upward’ trend. In the core with a permeability of 150 × 10^−3^ μm^2^, the polymer solution also exhibited viscoelasticity when the shear rate reached 1000 s^−1^ and the polymer viscosity exceeded 125 mPa·s, and the slope of the viscosity curve was greater than that of the core with a permeability of 100 × 10^−3^ μm^2^. Based on the above results, under the current polymer-type conditions, when the polymer viscosity reached 125 mPa·s, the equivalent shear rate reached 1000 s^−1^. When the effective permeability of the core reached 100 × 10^−3^ μm^2^, the polymer solution exhibited an elastic effect in the porous medium [48].

### 3.2. Force Analysis of Microscopic Remaining Oil

#### 3.2.1. Normal Deviator Stress on Microscopic Remaining Oil

To analyse the microscopic forces acting on various polymers passing through the remaining oil, PolyFlow software (ANSYS, Inc., Canonsburg, PA, USA) was used to calculate the pressure gradient in microchannels of 0.02 MPa/m, Newtonian fluid (oil) with a viscosity of 10 mPa·s, and polymers with different viscosities. The stress field was calculated for the flow of the elastic fluid through the remaining oil, as shown in Figure 8. In this study, the normal stress was positive and negative, the pressure was positive, and the tension was negative. The tangential stress was positive along the flow direction and negative against the flow direction. Horizontal and vertical stress was positive in the coordinate axis direction, and negative in the opposite direction. As the viscosity of the displacement fluid increased, the stress acting on the remaining oil also increased. When the viscosity doubled, the stress on the remaining oil almost doubled. Normal stress is the force perpendicular to the surface of the remaining oil [49]. This force causes the remaining oil to bulge or sag in the normal direction, which increases its local deformation. The remaining upstream oil is compressed, and the remaining downstream oil is stretched. Tangential deviator stress is the force along the tangential direction of the residual oil. The greater the force, the greater the angular deformation. Tangential stress is the greatest in the middle of the remaining oil, illustrating an inverted parabolic distribution law. The horizontal stress describes the changing law of stress along the direction of flow. The horizontal stress experienced by the remaining oil is positive, and thus, it deforms along the direction of flow. Upstream of the remaining oil, the vertical stress is downward, and the downstream vertical stress is upward, also reflecting the deformation process of the remaining oil after it is stressed.

#### 3.2.2. Horizontal Stress Differences on Microscopic Remaining Oil

The above calculation results for different types of stress only present the difference in the stress distribution of the remaining oil under different viscosities and cannot intuitively provide the difference in the displacement force acting on the remaining oil. The force that truly reflects the displacement effect of the displacement fluid on the remaining oil is the difference in the horizontal stress acting on the same horizontal line of the remaining oil interface, that is, the horizontal stress difference, as shown in Figure 9. This figure presents the change in the horizontal stress experienced by the remaining oil under the action of displacement fluids with different viscosities. The horizontal stress difference reflects the magnitude of stress of the displacement fluid acting on the remaining oil at different positions along the flow direction. When the viscosity of the displacement fluid increased, the horizontal stress difference also increased. When the viscosity doubled, the horizontal stress difference also doubled. When displacement fluids with different viscosities flowed through the remaining oil, shearing and stretching forces were generated, such that the distribution of remaining oil changed, and the wetting angle also changed accordingly, resulting in wetting hysteresis. Along the flow direction, the rear part of the remaining oil was pushed downstream towards the wall, producing a negative curvature, while the front part of the remaining oil tended to roll along the wall. If the interfacial tension between the rock and remaining oil was high, the remaining oil would eventually break, with part of it separating from the mother liquid droplet, and part of it continuing to adhere to the wall [50,51].

### 3.3. Microscopic Visualization of Oil Displacement

#### 3.3.1. Influence of Polymer Viscosity on the Microscopic Oil Displacement Effect

The arrows in the microscopic visualisation model in Figure 10 represent the displacement direction. Polymer solutions with different viscosities were prepared for the micro-displacement experiments. After water flooding, the total amount of residual oil significantly decreased. However, a large amount of residual oil remained in the model. After waterflooding, polymer flooding was performed using polymers with different viscosities, and the large-area clusters retained after waterflooding were divided into multiple small-area clusters of the remaining oil. As the viscosity of the polymer solution increased, the recovery factor of the model continued to increase. When the viscosity of the polymer reached 215 mPa·s, the recovery factor in the microscopic visualisation model was high, but a large amount of microscopic oil was still retained.

A microscopic visual image analysis system was used to calculate the recovery factor during different displacement stages, and the results are listed in Table 2. As the viscosity of the polymer increased, its viscoelasticity became more obvious, and its contribution to the oil displacement efficiency increased significantly. The microscopic remaining oil production in the model increased and the recovery factor increased. When the polymer viscosity was 215 mPa·s, the final recovery factor was 71.682%, which was 40.61% higher than that during waterflooding, and 28.318% of the remaining oil in the model could not be removed.

#### 3.3.2. Microscopic Remaining Oil Production at Different Locations

To specifically analyse the utilisation of polymers with different viscosities on the microscopic remaining oil at different locations, the visualisation model was divided into nine regions, and the nine regions were further divided into three types: main flow lines, diversion lines, and microscopic unaffected areas. The main flow line included Areas 3, 5, and 7, and the diversion line included Areas 2, 4, 6, and 8. The microscopically unaffected areas included Areas 1 and 9. The oil recovery factor for different area types was then calculated, as shown in Figure 11.

1.Mainstream area.

A comparative analysis of the production of microscopic residual oil in the main flow line during flooding with polymers with different viscosities is shown in Figure 12. Area 7 represents the injection end. When the displacement phase viscosity was low, the displacement pressure difference was small, the fluidity control ability was weak, and the driving force was less than the seepage resistance. The injected water protruded toward the production end along the dominant seepage channel and advanced forward in a non-piston form in the model. Fingering and circumvention pose serious problems; in this area, many clusters of residual oil formed because of circumvention and other reasons. As the viscosity of the displacement phase increased gradually, the viscoelasticity increased, resistance increased during the seepage process, the displacement pressure difference increased, and the spread range expanded. The partially retained clusters of the remaining oil changed from large clusters to scattered small clusters. When the polymer viscosity reached 215 mPa·s, the remaining oil volume in the microscopic visualisation model significantly decreased. However, as the seepage distance increased, the polymer fluidity control ability weakened, and bypass and fingering still occurred. The remaining oil volume in the main flow line from the injection end to the production end gradually increased.

A microscopic visualisation oil displacement image analysis system was used to calculate the remaining oil area in different areas on the mainstream line, and the recovery factor at different stages was calculated through the changes in the microscopic remaining oil in the field of view. The results are presented in Table 3. In the main flow line area, under the same viscosity conditions, as the seepage distance increased, the increase in oil recovery gradually decreased from the injection end to the production end. In the same area, as the viscosity of the polymer increased, the viscoelastic effect enhanced and the recovery factor gradually improved.

2.Diversion line area.

After flooding with polymers with different viscosities, the remaining oil in the model had a significant difference in the displacement effect on the mainstream line, and the distribution characteristics of the microscopic remaining oil on the diversion line were also significantly different. The remaining oil production at the injection end, Areas 8 and 4, and at the production end, Areas 6 and 2, were compared and analysed, as shown in Figure 13. When the polymer viscosity was 40 mPa·s, owing to the low viscosity of the displacement phase and the small driving force, the solution migrated along large pores with little resistance against forming a dominant seepage channel. As the injection volume increased, the remaining oil in the diversion line area no longer changed. As the viscosity of the displacement phase increased, the range of spread increased, effectively suppressing the fingering phenomenon. Not only did the remaining oil production on the mainstream line increase but the remaining oil on the diversion line also significantly decreased. After polymer flooding, oil still remained in the diversion line area, and the potential to tap this microscopic remaining oil was still relatively high.

A microscopic visual oil displacement image analysis system was used to calculate changes in the recovery factor in different areas of the shunt line. The results are presented in Table 4. As Areas 8 and 4 were closer to the injection end, the recovery factor was higher. However, owing to the differences in the range of polymer spread, the recovery factor in the shunt line area was significantly lower than that in the mainstream area.

3.Microscopic unaffected area.

In the microscopic visualisation model area division, Areas 1 and 9 are at the edges of the model, and the remaining oil in these areas is difficult to produce, as shown in Figure 14. When the viscosity was low, the displacement fluid had a lower degree of production for the remaining oil in these two areas, mainly as these two areas were perpendicular to the displacement direction, and the displacement force in the seepage direction could not effectively spread to these areas. As a result, the remaining oil in Areas 1 and 9 near the edges of the model was not extracted, and only a small amount of the remaining oil was carried away near the edge of the diversion line area. As the viscosity of the polymer gradually increased, the amount of oil remaining in the microscopically unaffected area significantly decreased. The main reason for this is that, as the viscosity of the displacement phase increases, the resistance increases during the seepage process, the displacement pressure difference increases, and the scope of the polymer solution significantly expands during the displacement process. The viscoelastic effect of the polymer solution can increase its seepage capacity in the porous medium and the production of remaining oil from these two areas. The residual oil in the unaffected microscopic area at the edge of the model thus significantly decreased.

A microscopic visual oil displacement image analysis system was used to calculate the recovery factor for different viscosities in the unaffected microscopic area. The results are presented in Table 5. The proportion of remaining oil in the microscopically unaffected area after flooding with polymers with different viscosities decreased. When the polymer viscosity was 215 mPa·s, the recovery factors were only 21.05% and 20.31%, indicating that the remaining oil in the microscopic unaffected area was relatively low and that there was still a large amount of retained oil.

## 4. Conclusions

(1)As per the creep recovery experiment, the greater the viscosity of the polymer solution, the greater the proportion of elasticity. Atomic force microscope images demonstrated that when the polymer concentration was low, the particles were almost isolated and did not form an ordered distribution, resulting in a low viscoelasticity of the solution. As the polymer concentration increased, the distance between particles decreased and aggregates formed. Nevertheless, the distribution of aggregates was not uniform, and the distance between them was relatively large. When the viscosity increased to 125 mPa·s, an orderly arrangement of polymer chain segments formed, and a relatively regular network structure formed between particles. The structure of the solution was ideal at this point, and the proportion of elasticity reached its maximum.(2)When the polymer viscosity reached 125 mPa·s, the equivalent shear rate reached 1000 s^−1^, the core effective permeability reached 100×10^−3^ μm^2^, the polymer solution exhibited viscoelasticity in the porous medium, and the viscosity curve exhibited an ‘upward’ phenomenon.(3)Based on the calculations from the numerical simulations using PolyFlow software, as the viscosity of the polymer increased, the stress acting on the remaining oil increased. When the viscosity doubled, the stress on the oil film also doubled. As displacement fluids of different viscosities flowed through the remaining oil, differences occurred in the normal and horizontal stresses, which deformed the remaining oil and changed the wetting angle accordingly.(4)In the microscopic visualisation flooding experiment, as the viscosity of the polymer increased, the degree of production of the remaining oil in the model gradually increased. Under the same viscosity conditions on the main flow line, as the seepage distance increased, the recovery factor gradually decreased from the injection to the production end. In the same area, as the viscosity of the polymer increased, the recovery factor gradually increased. Increasing the viscosity of the polymer significantly increased the sweep range in the shunt line area, but the recovery factor was significantly lower than that of the mainstream line. Increasing the viscosity of the polymer also affected the remaining oil in the microscopic unaffected area. When the polymer viscosity was 215 mPa·s, the recovery factors were only 21.05% and 20.31%, indicating that the remaining oil in the microscopic unaffected area after polymer flooding was relatively low, with a substantial amount of oil still being retained.

## Figures and Tables

**Figure 1 polymers-14-00940-f001:**
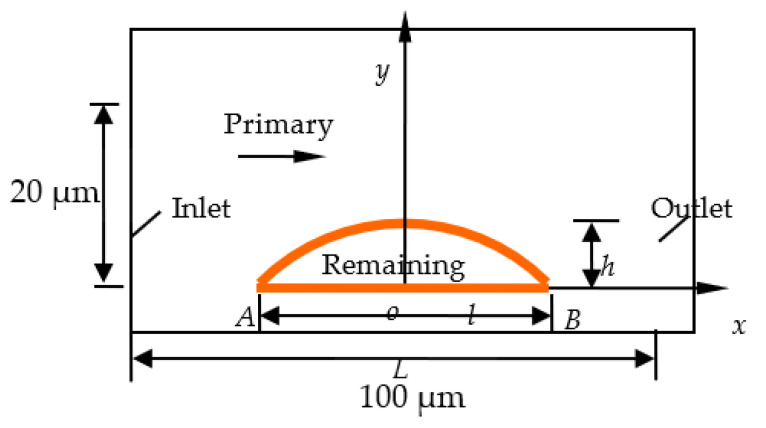
Calculation area diagram.

**Figure 2 polymers-14-00940-f002:**
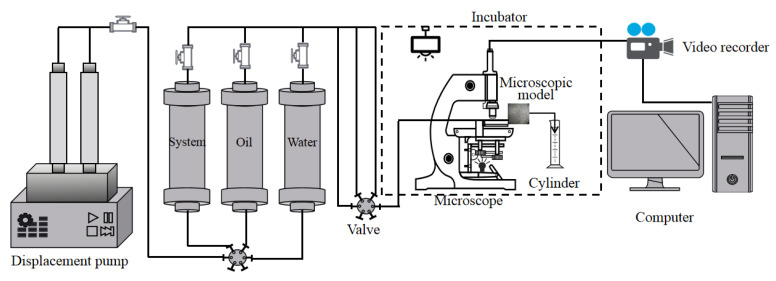
Microscopic visualisation oil displacement device.

**Figure 3 polymers-14-00940-f003:**
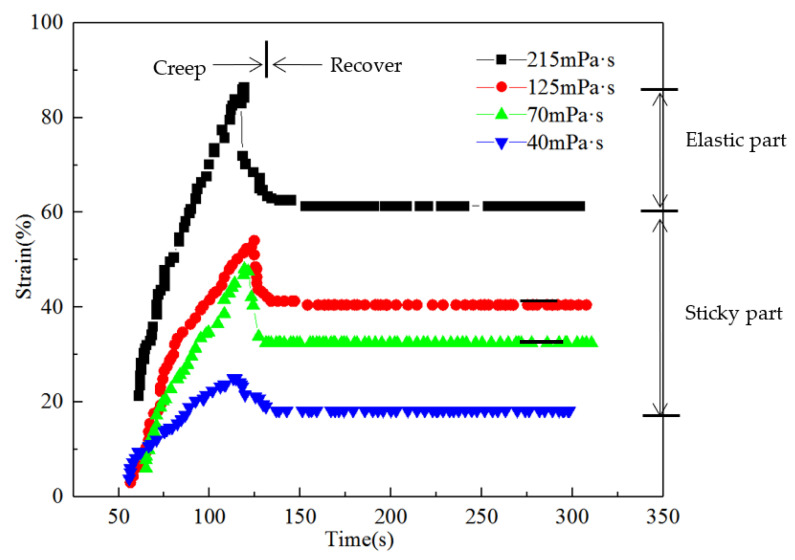
Creep recovery curves of polymer solutions with different viscosities.

**Figure 4 polymers-14-00940-f004:**
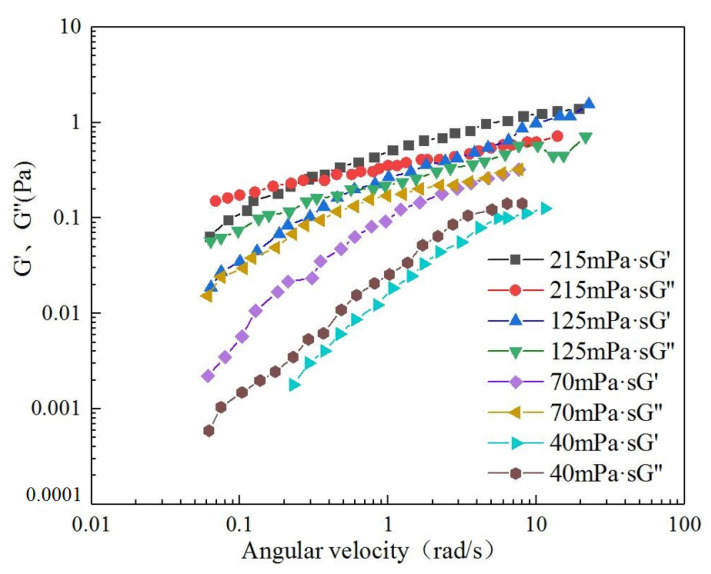
Changes in the storage and energy dissipation moduli of polymers with different viscosities.

**Figure 5 polymers-14-00940-f005:**
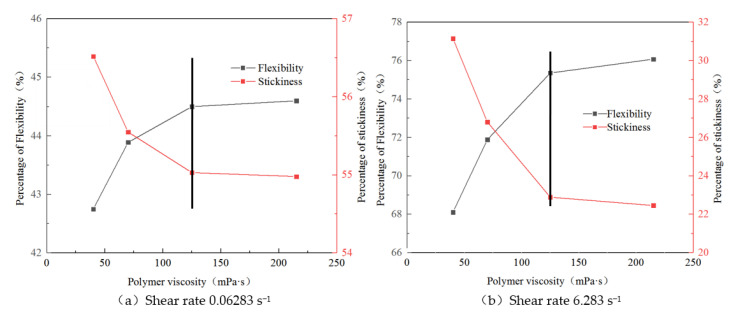
Proportion of viscoelasticity under different shear rates and concentrations. (**a**) Shear rate 0.06283 s^−1^. (**b**) Shear rate 6.283 s^−1^.

**Figure 6 polymers-14-00940-f006:**
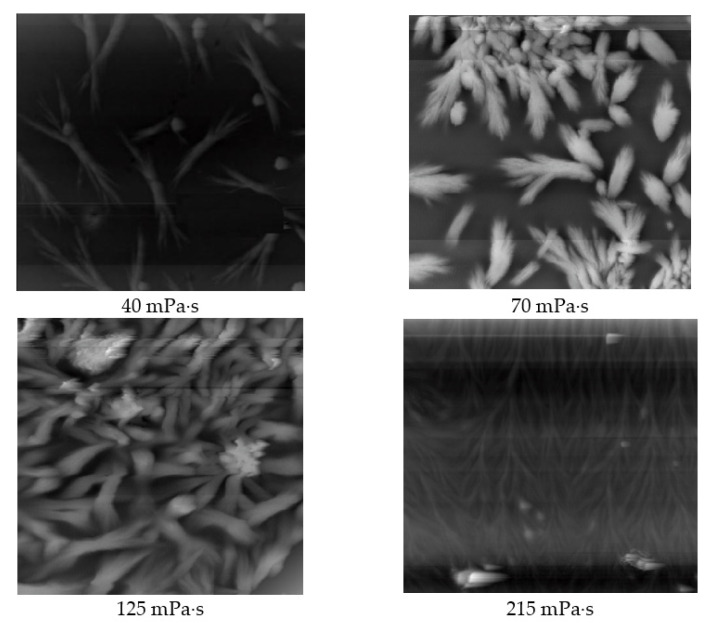
Structure and morphology of polymer solutions with different viscosities (20 μm × 20 μm).

**Figure 7 polymers-14-00940-f007:**
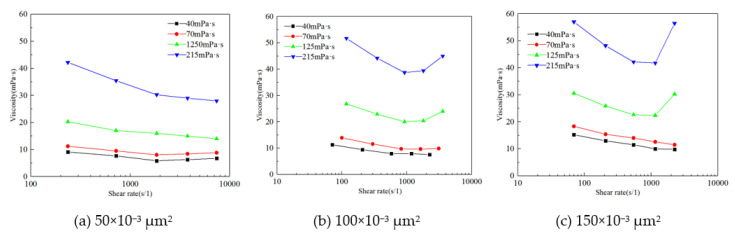
Core viscosity changes at different permeabilities. (**a**) 50 × 10^−3^ μm^2^. (**b**) 1.00 × 10^−3^ μm^2^. (**c**) 1.50 × 10^−3^ μm^2^.

**Figure 8 polymers-14-00940-f008:**
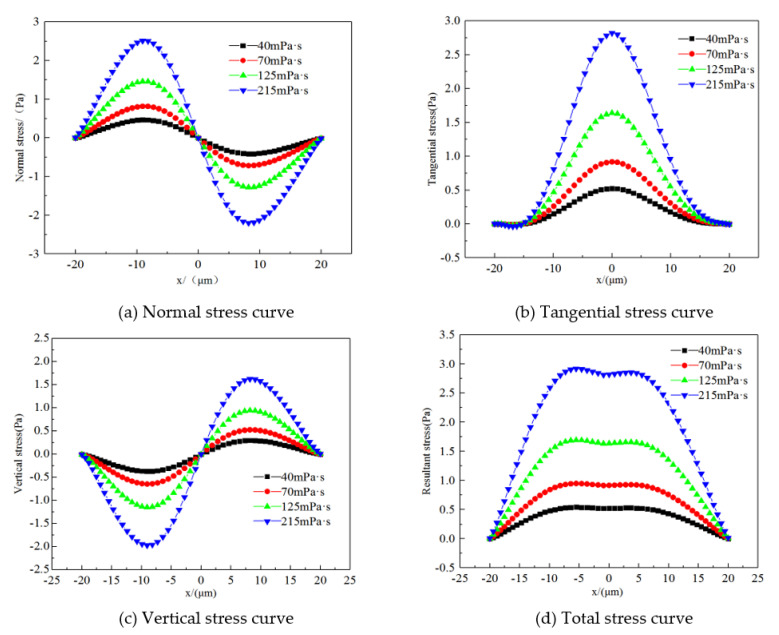
Stress curves. (**a**) Normal stress curve. (**b**) Tangential stress curve. (**c**) Vertical stress curve. (**d**) Total stress curve.

**Figure 9 polymers-14-00940-f009:**
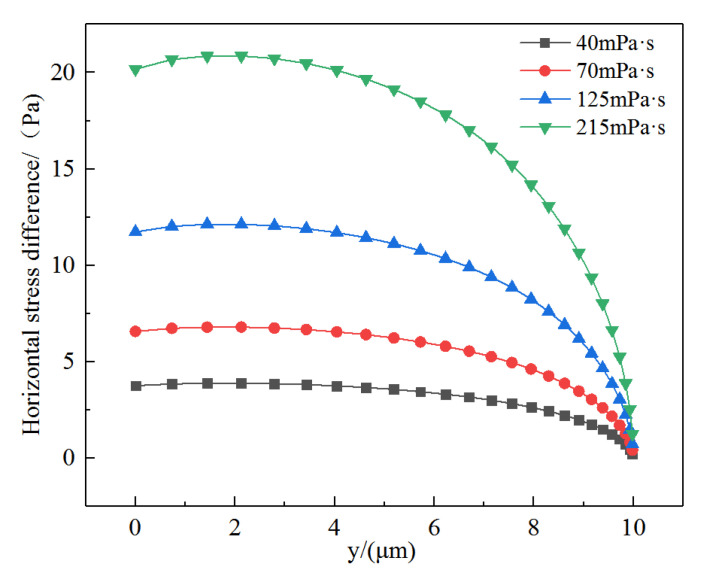
Variation of horizontal stress differences.

**Figure 10 polymers-14-00940-f010:**
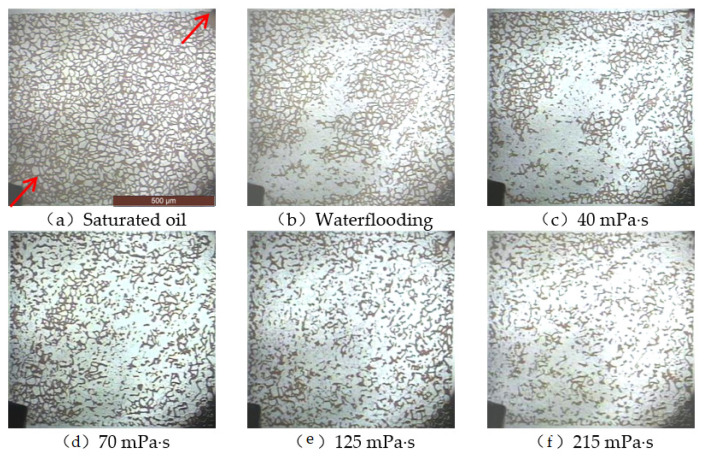
Oil displacement effect of polymers with different viscosities. (**a**) Saturated oil. (**b**) Waterflooding. (**c**) 40 mPa·s. (**d**) 70 mPa·s. (**e**) 125 mPa·s. (**f**) 215 mPa·s.

**Figure 11 polymers-14-00940-f011:**
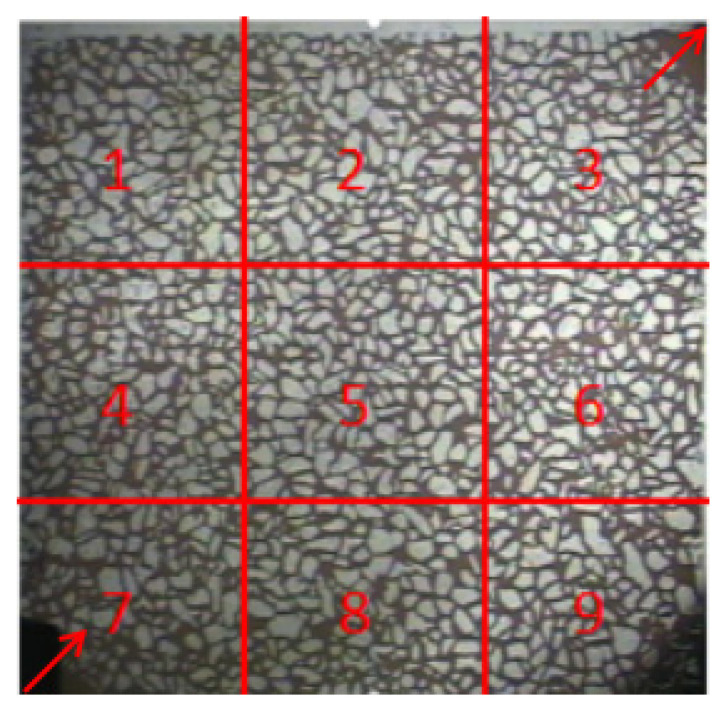
Regional division of the microscopic visualisation model.

**Figure 12 polymers-14-00940-f012:**
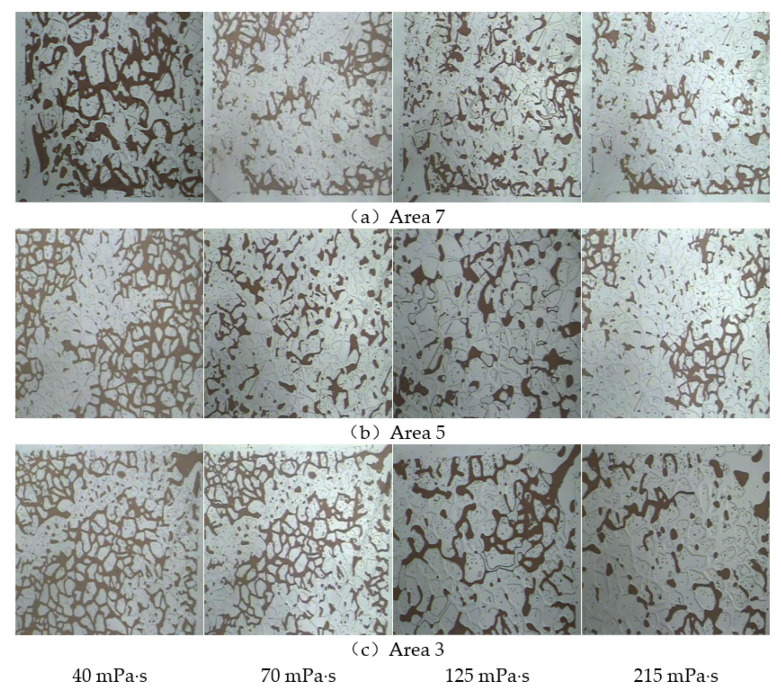
Microscopic remaining oil production in the mainstream area under different viscosity conditions. (**a**) Area 7. (**b**) Area 5. (**c**) Area 3.

**Figure 13 polymers-14-00940-f013:**
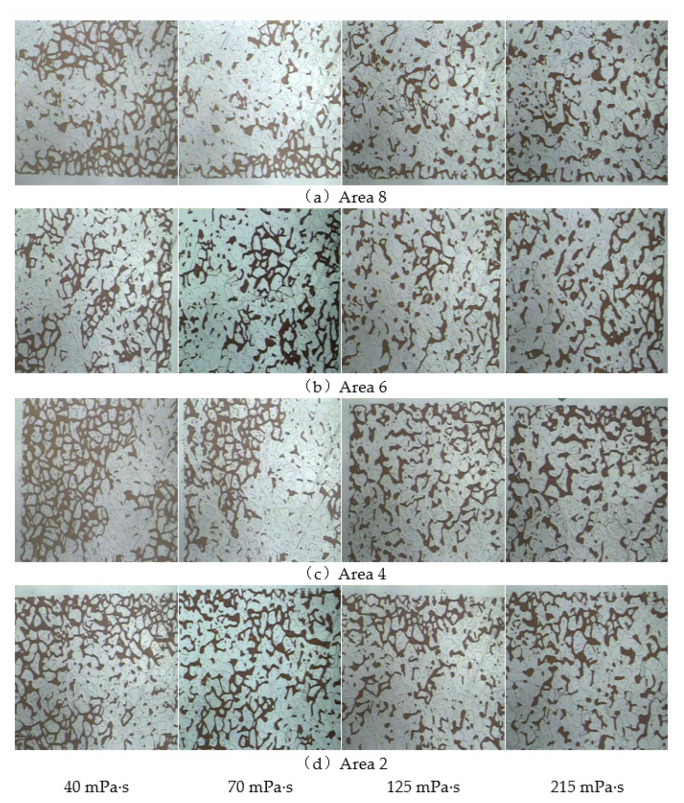
Microscopic remaining oil production in the shunt line area under different viscosity conditions. (**a**) Area 8. (**b**) Area 6. (**c**) Area 4. (**d**) Area 2.

**Figure 14 polymers-14-00940-f014:**
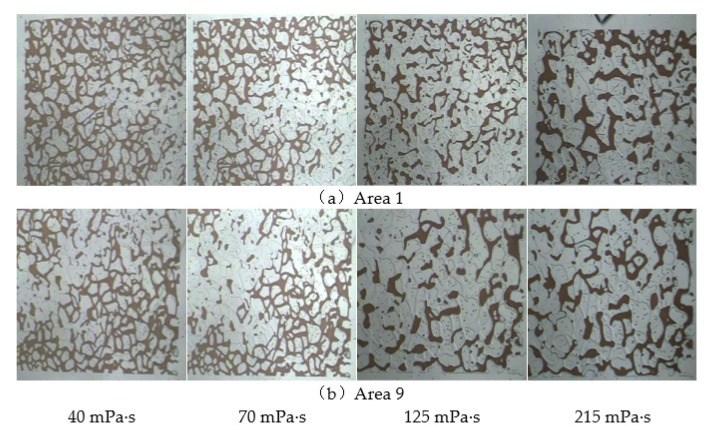
Distribution characteristics of remaining oil in the microscopic unaffected area after polymer flooding. (**a**) Area 1. (**b**) Area 9.

**Table 1 polymers-14-00940-t001:** Viscosity of polymer solutions with different molecular weights and concentrations.

Polymer Molecular Weight	Polymer Concentration (mg/L)	Viscosity (mPa·s)
25 million	870	40
25 million	1500	70
25 million	2000	125
25 million	2500	215

**Table 2 polymers-14-00940-t002:** Recovery efficiency from flooding with polymer with different viscosities.

Viscosity (mPa·s)	Polymer Flooding Recovery Factor (%)	Ultimate Recovery (%)	Enhanced Value of Polymer Flooding Recovery Factor (%)
Water drive	31.072	31.072	-
40	58.503	58.503	27.431
70	59.675	65.563	34.491
125	58.663	66.996	35.924
215	60.353	71.682	40.610

**Table 3 polymers-14-00940-t003:** Changes in recovery factors in different areas of the mainstream line.

Position	Recovery Efficiency of Polymer Flooding at Different Viscosities (%)			
	40 mPa·s	70 mPa·s	125 mPa·s	215 mPa·s
Area 7	51.25	59.64	68.26	72.23
Area 5	44.20	58.04	64.05	70.19
Area 3	37.02	50.78	59.65	63.84

**Table 4 polymers-14-00940-t004:** Changes in the recovery factor in different areas of the diversion line.

Position	Recovery Efficiency of Polymer Flooding with Different Viscosities (%)			
	40 mPa·s	70 mPa·s	125 mPa·s	215 mPa·s
Area 8	23.17	25.95	35.16	41.19
Area 6	15.73	25.40	34.52	45.28
Area 4	22.35	24.50	29.42	37.50
Area 2	11.57	16.58	20.73	25.87

**Table 5 polymers-14-00940-t005:** Variations in recovery in the microscopic unaffected area.

Position	Recovery Efficiency of Polymer Flooding with Different Viscosities (%)			
	40 mPa·s	70 mPa·s	125 mPa·s	215 mPa·s
Area 1	25.66	24.12	23.75	21.05
Area 9	24.03	23.09	22.70	20.31

## Data Availability

Not applicable.

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
