# Peer review of "Influence of Polymer Viscoelasticity on Microscopic Remaining Oil Production"

_polymers, 2022, doi:10.3390/polym14050940_

Round 1

Reviewer 1 Report

In this manuscript, the authors reported the influence of polymer viscoelasticity on microscopic remaining oil production by applying the microscopic visualization oil displacement technology combined with numerical simulation technology and an indoor core seepage experiment.  I would recommend the acceptance of this manuscript after the following revision:

  1. Figure 3, The creep recovery curve for the sample with 70mPa·s viscosity shows a hump in the recovery phase, can the authors explain a little bit more why that is?
  2. Figure 4, the slope of the G' (G") versus angular velocity curve decreases with increasing the viscosity of the polymer solution, can the authors explain the reason behind this?
  3. Figure 6, the AFM experiment should be mentioned in the experimental methodology section. It seems like the size of the polymer aggregates increases with increasing the solution viscosity from 40 to 125 mPa·s, but the polymer aggregates for these three samples have similar morphology. However, for the 215mPa·s sample, a completely different morphology was observed. Can the authors expand the discussion on this point?
  4. Can the authors add a scale bar to Figure 10?
  5. Table 2, please write your manuscript in English.
  6. Figure 11, are the arrows placed at the correct locations?
  7. Table 3-5, the recovery efficiency enhancement from 40 mPa·s to 215 mPa·s is different for different areas, is there a trend? which area has the greatest efficiency enhancement?
  8. Figure 13, why area 2 and 6 are not included?
  9. English expression should be carefully checked and revised throughout the manuscript. 

Author Response

Dear Reviewer

please see the document attached

Reviewer 2 Report

The paper under review reports on e the impact of polymer viscoelasticity on microscopic remaining oil production. In my opinion the paper can be published in Polymers after some minor corrections (according to the comments listed below):

Minor comments:

1) Line 129: please cancel "in the figure", as the dimensions given here simply refere to the studied system.

2) Line 140: please use the same symbols within the text and in equations (e.g. \rho)

3) Line 143: should be: "\rho=const."

4) Fig. 5: please use slighty thicker lines for flexibility and stickness

5) Line 299: in my opinion the authors should use here "rates" or "velocities" instead of "speeds".

6) Fig. 7: please use larger fonts

7) Table: first entry in the column "viscosity" needs correction

8) Line 400: starting from this line there is something wrong with the numeration of sections/subsections.

Author Response

(The authors gave the same response as above.)

Round 2

Reviewer 1 Report

The authors have addressed my concerns, I recommend the acceptance of the manuscript in the present form. 

Author Response

Thanks very much for spending time reading our manuscript and coming up with valuable comments and suggestions.